# Ultrasound-Assisted Demineralization Process of Sugarcane Straw and Its Influence on the Further Biomass Conversion

Daniel Santos, Karen Giacobe, Carolina M. Silva, Loisleini F. Saldanha, Ayrton F. Martins, Erico M. M. Flores and Cezar A. Bizzi *

Department of Chemistry, Federal University of Santa Maria, Santa Maria 97105-900, RS, Brazil; danielchemistryeng@gmail.com (D.S.); karengiacobe@gmail.com (K.G.); carolmarquezan@gmail.com (C.M.S.); loissaldanha@gmail.com (L.F.S.); ayrtonfm@gmail.com (A.F.M.); ericommf@gmail.com (E.M.M.F.)
* Correspondence: cezar.bizzi@ufsm.br

**Abstract:** Lignocellulosic materials have been considered as an alternative source from which liquid biofuel and fine chemicals can be produced with a moderate environmental impact. However, they can be contaminated with metals, soil, and ash, owing incrustation and corrosion of industrial reactors and pipelines. In this work, the use of ultrasound energy was applied for the removal of metals and nonmetals (Ba, Ca, Mg, Mn, P, S, Si, and Sr) from sugarcane straw. Ultrasound-assisted demineralization (UAD) experiments were carried out in ultrasonic baths in several frequencies (from 25 up to 130 kHz). The following experimental conditions were evaluated: demineralization solution ($HNO_3$, $HCl$, $H_2SO_4$, $H_2O_2$, and $H_2O$), $H_2O_2$ concentration (from 5 to 30% v $v^{-1}$), extraction temperature (from 30 to 70 °C), sonication time (from 5 to 45 min), and ultrasound amplitude (from 10 to 70%). Better demineralization efficiencies (66%) were obtained employing an ultrasound bath operating at 25 kHz for 30 min, ultrasound amplitude of 60%, and using a diluted $H_2O_2$ solution (15% v $v^{-1}$) at 70 °C. When the obtained results were compared with those obtained by mechanical stirring (MS, 500 rpm), it was observed that the use of ultrasound energy increased the demineralization efficiency up to 16%. Furthermore, acid hydrolysis was performed to evaluate the influence of US and mechanical stirring in fermentable sugars' production. The total sugars' yield (glucose, xylose, and arabinose) increased around 55% for both systems (US and MS). To prove the applicability of the proposed process, some experiments for scaling up were performed using several reaction loads (0.5 to 3 L). An attempt for scaling the proposed process up was well succeeded up to a 3 L load. Therefore, the proposed ultrasound-assisted procedure can be considered as a suitable alternative for high-efficiency demineralization from sugarcane straw.

**Keywords:** demineralization; waste valorization; biomass conversion; acoustic cavitation

## 1. Introduction

The decrease of fossil fuels resources associated with the need for the reduction of pollution and greenhouse gas emissions makes lignocellulosic biomass a promising alternative source for fuel and solvent production. The use of biomass leads to a significant reduction of the environmental impact associated with energy production [1,2]. Lignocellulosic materials trap atmospheric carbon dioxide, reducing carbon and greenhouse gas emissions [3].

Considering any process regarding biomass valorization, a pretreatment step is necessary to ensure the selectivity and efficient biomass conversion. The pretreatment step consists of the fragmentation of lignocellulosic material, and consequently separation of lignin, hemicellulose, cellulose, and inorganic materials [4]. Recently, potentially alternative feedstocks to food raw materials have been studied. The so-called "second generation feedstocks" are considered wastes or subproducts of conventional routes, such as food industry waste, agricultural waste, peels, and straws [5]. Among the second-generation

feedstocks, sugarcane straw arises as a promising alternative to the biorefinery activities. Sugarcane straw is composed mainly of cellulose, hemicellulose, and lignin (approximately 45, 28, and 18% wt., respectively), which ensure versatility in the use of this feedstock in industrial activities [6–8].

The problem associated with lignocellulosic material considered as waste is related to the presence of metals and nonmetals (lignocellulosic structure or contamination), soil and ash (contamination from harvest process) [9,10]. These are the main reasons that up to a few years ago, these residues were not of industrial interest since they represent a source of incrustation and corrosion of industrial reactors and pipelines. Moreover, the presence of inorganic materials are considered a critical issue in thermochemical processes owing to the reduction of available calorific energy for the industrial process [11]. Thus, a previous-step of demineralization, used for mineral removal, is required to allow further industrial uses, such as the sugarcane straw for generating energy and value-added products.

The conventional approaches to biomass demineralization are performed using diluted acid solutions with long extraction periods, at relatively high temperatures. Despite allowing contaminant removal, these pretreatments can degrade the biomass in a significant way. Alternatively, some studies reported the use of supercritical fluid, high mechanical stirring, and soxhlet extraction, which improve demineralization efficiency but become unviable for industrial activities [1–4].

Among the technologies employed for demineralization process intensification, ultrasound (US) energy has been reported as a promising alternative aimed at extraction approaches. US energy increases the mass transfer of the biomass minerals to the extraction solution and in specific conditions can lead to acoustic cavitation [12]. US waves propagates in alternate cycles of expansion and compression. During the rarefaction cycle, the cohesive forces of the liquid medium can be overcome, and a cavity of gas or vapor is formed (microbubbles). Considering specific conditions of US frequency and energy, as well as the liquid properties, these microbubbles reach a critical size and collapse. The cycle of formation, growth, and collapse of microbubbles under US field is called acoustic cavitation, responsible for releasing a high amount of energy in the form of shock waves, erosion, radical formation, and microjets that promote the demineralization of biomass [12–14]. Some studies reported synergic effects between US energy and extraction solvent for elemental removal. In these works, the main US effects to extract elements from biomass are associated with sonoporation, sonocapilarity, erosion, fragmentation, and deterioration of a solid matrix [15].

In this sense, the development of alternative pretreatments for metals' removal from lignocellulosic materials is considered an important research topic. In this work, the use of US energy to extract Ba, Ca, Mg, Mn, P, S, Si, and Sr from sugarcane straw was investigated aiming at biomass demineralization. These elements were selected as a proof of concept to assess the ultrasound-assisted demineralization (UAD) process. The following parameters for UAD were evaluated: US bath system operating at different frequencies and delivered power, and demineralization solution. Experiments using mechanical stirring (MS) as a silent condition were performed to compare the efficiency of the proposed UAD process. Additional experiments using an ultrasonic reactor for process scaling were investigated. Finally, the untreated sugarcane straw was compared with the one treated with US to evaluate the yields of fermentable sugars obtained by acid hydrolysis, which gave information about the feasibility of the use of biomass in biorefinery activities.

## 2. Materials and Methods

### 2.1. Instrumentation

Ultrasound bath systems were evaluated operating at different frequencies: 25 or 45 kHz with nominal power of 100 W (Transsonic TI-H-5 3.5 L, Elma GmbH & Co., Singen, Germany); 37 or 80 kHz with nominal power of 330 W (Transsonic P120H 9.0 L or P120H 12.9 L, Elma GmbH & Co., Singen, Germany); 35 or 130 kHz with nominal power of 200 W (Transsonic TI-H-10 8.6 L, Elma GmbH & Co., Singen, Germany).

Experiments for scaling up were performed using an ultrasonic reactor (stainless steel bowl with 240 mm of length, 130 mm of width, and 150 mm of height) operating at 25 kHz with a capability of up to 3.5 L.

To compare the UAD process, experiments in silent conditions (without US) were performed using a mechanical stirrer (model PT 3100 D, Polytron, Kinematica, Switzerland) equipped with a homogenizer bar (20 mm of diameter) operating at 500 rpm under same experimental conditions.

For all experiments, metals and nonmetals were determined by inductively coupled plasma optical emission spectrometry (Optima 4300 DV, Perkin Elmer, Waltham, MA, USA) using an axial view mode, equipped with a cyclonic spray chamber and a GemCone nebulizer. Before element determination, biomass was digested using a microwave-assisted sample preparation system (SpeedwaveTM Four, Berghof, Eningen, Germany).

Sugarcane straws with and without pretreatments were characterized by scanning electron microscopy (model Sigma 300 VP, Carl Zeiss, Jena, Germany) operating in the high vacuum mode with a secondary electron detector for qualitative analysis of images.

In addition to the development of the UAD process, experiments of acid hydrolysis were performed using sugarcane straws with and without treatments to elucidate whether the ultrasound could significantly affect sugar production. Experiments were performed using an Autoclave (model CS, Prismatec, Porto Alegre, Brazil) for 30 min and with 15 lb inc$^{-2}$. The production of glucose, xylose, and arabinose was determined by high performance liquid chromatography coupled to a refractive index detector (HPLC-RID, Shimadzu, Tokyo, Japan).

### 2.2. Reagents

The distilled water used to prepare all standard solutions and reagents was purified using a model Milli-Q system (Millipore Corp., Burlington, VT, USA). All reagents used in this work were of analytical grade. The following reagents were used for demineralization, decomposition, and/or hydrolysis: nitric acid ($HNO_3$, Sigma-Aldrich, St. Louis, MO, USA), hydrogen peroxide ($H_2O_2$, Moderna, Cambridge, MA, USA), hydrochloric acid (HCl, Merck, Darmstadt, Germany), sulfuric acid ($H_2SO_4$, Sigma-Aldrich), and hydrofluoric acid (HF, Sigma-Aldrich). Nitrogen and argon (purity of 99.999% and 99.998%, respectively, White Martins, Rio De Janeiro, Brazil) were used for SEM and ICP-OES analysis. In the development of the chromatographic method for the determination of the composition of the hydrolysates by HPLC-RID, anhydrous D-glucose (96%), xylose ($\geq$99%), and arabinose ($\geq$98%) from Sigma-Aldrich (St. Louis, MO, USA) were used. Acetonitrile and methanol used in HPLC analysis were purchased from Sigma (Roedermark, Germany) and Tedia Company (Fairfield, CT, USA), respectively. A multielement stock reference solution of 10 mg L$^{-1}$ (SCP33MS, SCP Science, Quebec, QC, Canada) was used for metal and nonmetal determination by ICP-OES.

### 2.3. Sugarcane Straw

Sugarcane straw used in this study was obtained from a local farm in Rio Grande do Sul State (Brazil). The sample was cut into small pieces with diameters lower than 2 mm. The elemental concentration in untreated sugarcane straw, obtained after sample digestion, was performed by ICP-OES, which was used to calculate the demineralization efficiency, as described in Section 2.4.

### 2.4. Methods

The demineralization process was carried out in a glass cylindrical reactor with a 5 cm internal diameter and 12 cm height. The ration of initial mass of sugarcane straw: volume of demineralization solution was arbitrarily selected to be 250 mg in 20 mL. Preliminary experiments were performed using an US bath operating at 25 kHz with 100 W of nominal power and 70% of US amplitude. The extraction time and temperature were set at 30 min and 70 °C, respectively. Several demineralization solutions were investigated: $HNO_3$,

$H_2O_2$, HCl, $H_2SO_4$, and $H_2O$. Further, other US frequencies varying from 35 to 130 kHz were also evaluated.

After selecting the appropriate setup for UAD (US system and extraction solvent), the following experimental parameters were evaluated: $H_2O_2$ concentration (from 5 to 30% v v$^{-1}$), extraction temperature (from 30 to 70 °C), sonication time (from 5 to 45 min), and US amplitude (from 10 to 70%). Experiments in silent conditions (without ultrasound) were performed with the same conditions employed in the UAD but using an MS set at 500 rpm. In addition, experiments for scaling up were performed using an ultrasonic reactor operating at 25 kHz with several reactor loads (from 0.5 to 3.0 L).

After UAD, the pretreated sugarcane straw was collected, dried, and submitted to acid digestion for further metal and nonmetal determination by ICP-OES. Microwave-assisted digestion of sugarcane straw was performed with a mixture of 5 mL of 14.4 mol L$^{-1}$ $HNO_3$ and 1 mL of 12 mol L$^{-1}$ HCl in the following heating program: (1) 10 min ramp up to 200 °C, (2) hold for 40 min at 200 °C. The sample mass used in the elemental characterization was 300 mg. For the sugarcane straw digestion step after demineralization, the sample was separated from the extracting solution, dried, and 200 mg of sample was digested. All the digested content in the vessels was diluted up to 20 mL with ultrapure water. For ICP-OES analysis, the solutions resulting from the sample digestion were diluted with $HNO_3$ 5% (same condition as the calibration curves) and injected into the equipment. Demineralization efficiency ($D_{ef}$) was calculated based on the results obtained with untreated sugarcane straw, as presented in Equation (1).

$$D_{ef} = \frac{[element]_{UAD}}{[element]_{total}} \times 100 \tag{1}$$

where $[element]_{UAD}$ is the element concentration determined in the pretreated sugarcane straw; $[element]_{total}$ is the element concentration determined in the in natura sugarcane straw.

Subsequent to optimization of the proposed process, the morphology and structure of untreated and pretreated biomass were characterized by SEM. A scanning electron microscope (model Sigma 300 VP, Carl Zeiss, Cambridge, UK) with a field emission gun (FEG) of the Schottky type (tungsten filament coated with zirconium oxide), equipped with a Gemini column (Carl Zeiss) was used in the variable pressure (VP) mode. Images were obtained using a variable pressure secondary electron detector (VPSE).

Moreover, experiments of acid hydrolysis were performed using sugarcane straw with and without pretreatments. The hydrolytic experimental conditions were selected according to previous literature (500 mg of sugarcane straw; 8.3 mL $H_2SO_4$ 1.5%; autoclaved during 30 min at 121 °C) [16]. The concentration of glucose, xylose, and arabinose after the hydrolysis process was determined by high performance liquid chromatography (HPLC) coupled to a refractive index (RID) detector. The chromatograph (Shimadzu, São Paulo, Brazil) was equipped with a LC-20AT pump, a communication module CBM-20A, a DGU-20A5 degasser, and an autosampler SIL-20A that was used with the aid of LC Solution software. The separation was performed using an Aminex HPX-87H column (300 mm × 7.8 mm; 9.0 μm particles) at 50 °C, 0.5 mL min$^{-1}$ flow (mobile phase 5 mmol L$^{-1}$ $H_2SO_4$) in isocratic mode. Prior to chromatographic separation, the samples were diluted in ultrapure water (1:10 v v$^{-1}$) and subsequently passed through syringe filters (PTFE or MCE 13 mm × 0.22 μm). The injection volume was 20 μL. The chromatographic run time corresponding to the analyte signal was 10.7 min with glucose, 11.5 min with xylose, and 12.6 min with arabinose.

## 3. Results and Discussion

### 3.1. Influence of Extraction Solution for Demineralization Process

To better evaluate the effect of the extraction solution, as well as every evaluated parameter for the proposed UAD, the elemental concentration in untreated sugarcane straw obtained after sample digestion was tested by ICP-OES. The results are present Table 1.

**Table 1.** Metals content from untreated sugarcane straw after MAWD (determination by ICP-OES, n = 3).

| Analytes | Concentration, $\mu g\ g^{-1}$ |
| --- | --- |
| Ba | $190 \pm 15$ |
| Ca | $5960 \pm 20$ |
| Mg | $1740 \pm 10$ |
| Mn | $760 \pm 40$ |
| P | $825 \pm 10$ |
| S | $1070 \pm 50$ |
| Si | $400 \pm 20$ |
| Sr | $45 \pm 1$ |

Although the extraction solution is decisive for the efficiency of demineralization, it would change the biomass integrity, and affect further biomass conversion. The extraction solution can modify the lignocellulosic structure, making the further conversion process easy, or it might solubilize the sample, negatively affecting the yields. To understand how US energy would affect the biomass, thus interfering with its further use, several solutions ($HNO_3$, $HCl$, $H_2SO_4$, $H_2O_2$, and $H_2O$) were evaluated as extractant. The experiments were performed using 250 mg of sugarcane straw and 20 mL of 15% (v $v^{-1}$) extraction solution. The extraction process was carried out for 30 min at 70 °C and 70% of US amplitude. The results obtained for the evaluation of each solution are present in Figure 1.

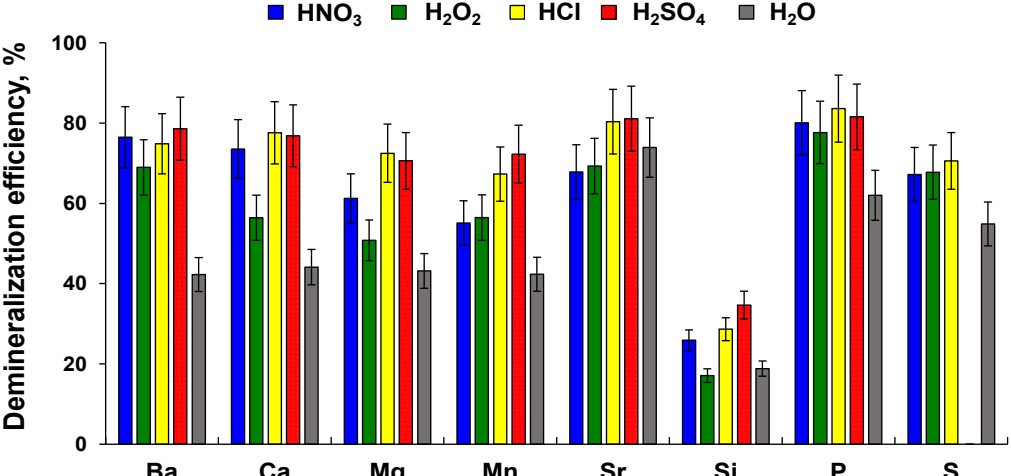

**Figure 1.** Demineralization efficiency with several extraction solutions using an ultrasonic bath operating at 25 kHz (100 W). Experimental conditions: 250 mg of sugarcane straw, 20 mL of 15% (v $v^{-1}$) extraction solution for 30 min at 70 °C, and 70% of US amplitude (error bars are the standard deviation, n = 3).

Except for Si, the extraction efficiency for the other evaluated elements ranged from 50 to 80% ($H_2O_2$, $H_2SO_4$, $HNO_3$, and $HCl$). Silicon, for instance, was extracted with very poor efficiency (from 17 to 35% for $H_2O_2$ and $H_2SO_4$, respectively). Silicon is considered a non-essential element for sugarcane and it is generally present in high concentrations throughout the plant [17]. According to the literature, Si is absorbed by the roots in the form of monosilicic acid and is deposited in plant tissues, polymerizing into phytoliths. Due to its amorphous structure and high hardness, the phytoliths contribute to the mechanical resistance and act as a defense mechanism of the sugarcane against microorganisms, extreme temperatures, and water deficit [9]. Due to the complexity of this structure, as a consequence of Si chemical bond and intermolecular forces, silicon cannot be efficiently extracted from sugarcane straw using the proposed reagents.

Considering all the evaluated elements, the extraction efficiency ranged from 48 to 72% when water or $H_2SO_4$ were used, respectively. Intermediate values of extraction efficiency were obtained for $H_2O_2$ (58%), $HNO_3$ (63%), and HCl (69%). Excluding water, the other extraction solutions were roughly equivalent (all solutions used at 15% v $v^{-1}$ concentration). Inorganic acids ($H_2SO_4$, $HNO_3$, and HCl) have been reported to be used as extraction solutions. However, the presence of high concentrations of S, N, and Cl in the pretreated biomass would represent problems related to catalyst poison, pipeline and reactor corrosion, and contamination/inactivation of the yeast used for further biomass conversion [18–20]. Hydrogen peroxide, for instance, becomes an interesting alternative since metal removal was satisfactory, and its decomposition results in water and oxygen [21,22]. In addition to generating a residue that is easy to manage and discard/reuse, the extreme conditions of pressure and temperature induced by acoustic cavitation in the presence of $H_2O_2$ may produce reactive and oxidant species such as hydroxyl, hydrogen, and hydroperoxyl radicals [23]. Although these reactive species improve the demineralization process, they further contribute to additional conversions of sugarcane straw. For these reasons, the $H_2O_2$ solution was selected for further UAD optimizations.

### 3.2. Evaluation of Different Ultrasound Systems

In view of differences among the energy provided from each US system, as well as to evaluate the influence of the US frequency from 25 to 130 kHz, ultrasound-assisted extraction was carried out using several US bath systems, as presented in Figure 2.

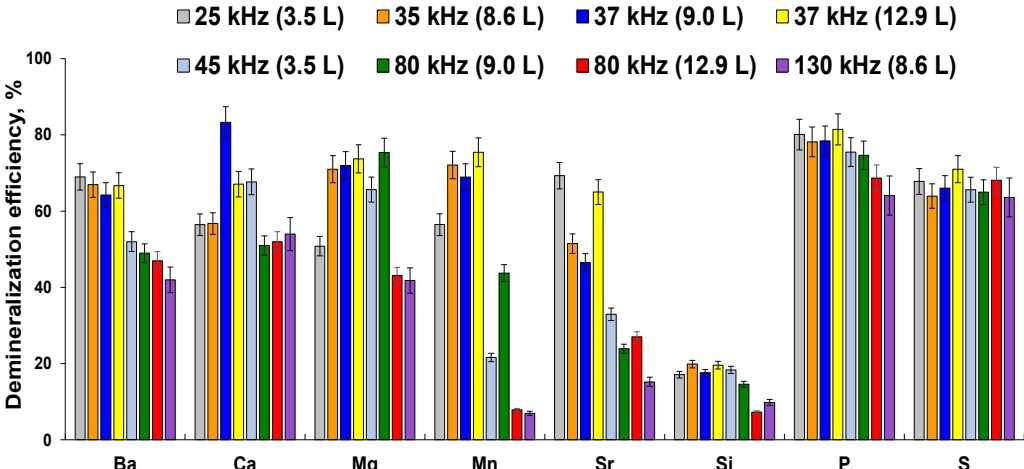

**Figure 2.** Demineralization efficiency with several US baths systems employing $H_2O_2$ as the extraction solvent. Experimental conditions: 250 mg of sugarcane straw and 20 mL of extraction solution 15% (v $v^{-1}$) for 30 min at 70 °C and 70% of US amplitude (error bars are the standard deviation, n = 3).

Baths are conventionally considered as an indirect application of ultrasonic energy because the wave is dissipated through the bath container filled with water before being absorbed by the reactor containing the reaction mixture (e.g., extraction solvent and biomass) [24]. Even so, satisfactory results of metal and nonmetal removal were obtained with the use of ultrasonic baths.

When a high US frequency from 45 to 130 kHz was used, extraction efficiencies from 37 to 46% were observed (45 and 130 kHz, respectively). When the same frequency was evaluated (80 kHz, internal volumes of 9.0 and 12.9 L), better extraction efficiencies were observed in a bath with a small volume container (filled with water), which is an indication that the power density is a parameter that contributes to metal and nonmetals extraction. In addition, degradation of the structure of the sugarcane straw was observed. This effect is undesirable since it results in the solubilization of the carbonaceous content of the biomass, resulting in the reduction of the organic matter available for further industrial uses, either

for conversion into fine chemicals or energy generation. As observed in Figure 2, ultrasonic systems operating from 25 to 37 kHz, for instance, presented better extraction efficiencies (from 58 to 65% for 25 and 37 kHz, respectively, for the pool of the elements). Unlike that observed in the system with frequencies higher than 45 kHz, the systems operating at frequencies ranging from 25 to 37 kHz were not so dependent on internal volume or the frequency. All the obtained results were very similar, and little solubilization of biomass was observed, which was closer to its original constitution.

Because of the low degradation of sugarcane straw, as well as the relatively high extraction efficiency, an US bath operating at 25 kHz with $H_2O_2$ as the extraction solution was selected as compromise conditions for further evaluations. In addition, it provides a relatively high power density (115 W dm$^{-3}$) released in the medium due to its low volumetric capacity (3.5 L).

### 3.3. Optimization of Demineralization Process

Although the US systems operating at lower US frequencies have presented better extraction efficiencies, there are other experimental parameters that must be carefully evaluated. Thus, the extraction solution ($H_2O_2$ from 5 to 30%), extraction temperature (from 30 to 70 °C), sonication time (from 10 to 40 min), and the US amplitude (from 10 to 70%) were evaluated, the last being related to the amount of US energy dissipated in the reaction medium.

The efficiency of the extraction solution was evaluated, and $H_2O_2$ was ranged from 5 to 30% (v v$^{-1}$). As observed in Figure 3, the extraction efficiency tended to increase from 5 to 15% $H_2O_2$ (from 23 to 57%, respectively, for the pool of the elements), where a kind of plateau was observed when more concentrated $H_2O_2$ was employed (around 56% of extraction efficiency). Thus, 15% $H_2O_2$ was selected as the extraction solution.

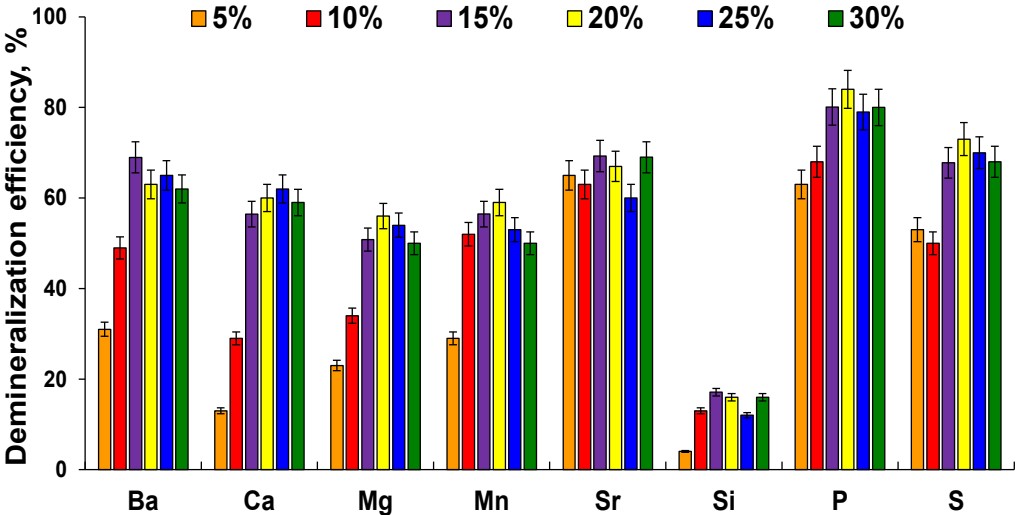

**Figure 3.** Demineralization efficiency with several $H_2O_2$ concentrations using an ultrasonic bath operating at 25 kHz. Experimental conditions: 250 mg of sugarcane straw and 20 mL of extraction solution for 30 min at 70 °C and 70% of US amplitude (error bars are the standard deviation, n = 3).

The extraction temperature for the UAD is another critical parameter to be evaluated because it influences the mineral leaching from the matrix. The extraction efficiency was evaluated ranging the temperature from 30 to 70 °C, as presented in Figure 4.

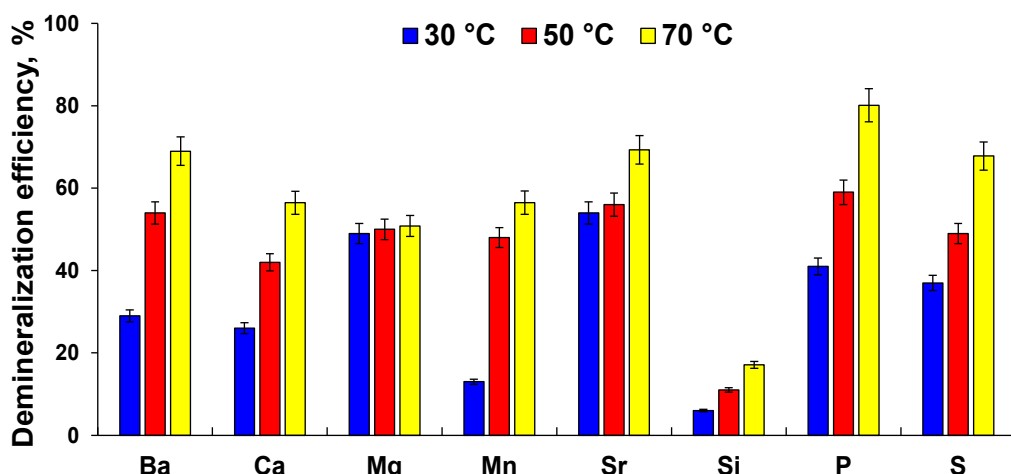

**Figure 4.** Demineralization efficiency with several extraction temperatures using an ultrasonic bath operating at 25 kHz. Experimental conditions: 250 mg of sugarcane straw and 20 mL of extraction solution 15% (v v$^{-1}$) for 30 min and 70% of US amplitude (error bars are the standard deviation, n = 3).

Generally, the higher the extraction temperature, the more efficient the extraction. However, when extraction is assisted by US, it is possible to work closer to room temperature, even using systems operating at atmospheric pressure. The extraction temperature affects the way US energy is delivered in the reaction medium, as well as the intensity associated with acoustic cavitation implosion [25]. At high temperatures, the energy released due to the implosion of the acoustic cavitation is damped. It happens because at higher temperatures, an increase of the solvent vapor pressure is observed, which reduces the collapse intensity [25,26]. As presented in Figure 4, there was a synergistic effect between the action of extraction solvent at higher temperatures and the propagation of the acoustic wave, which resulted in the higher extraction efficiency for the higher evaluated temperature (70 °C).

The sonication time and its influence on the efficiency of UAD of metals and nonmetals was evaluated. The time that a reactional mixture is exposed to US energy is an important parameter, mainly considering that US bath systems tend to deliver relatively low energy to the medium [27]. Thus, in order to reach the equilibrium where the elements are absorbed by the extraction solution, longer reaction times would be required [27]. As observed in Figure 5, there was a growing trend in elemental extraction from 10 to 30 min (from 30 to 58%, respectively, for the pool of elements). Practically the same extraction efficiency was observed for 30 and 40 min (58 and 60%), which means that there was no practical reason to keep the sonication over 30 min. Thus, considering that the extraction efficiency did not increase in a significant way, but 10 min of US operation certainly represents an extra cost for any industrial process, 30 min was selected as the optimal sonication time.

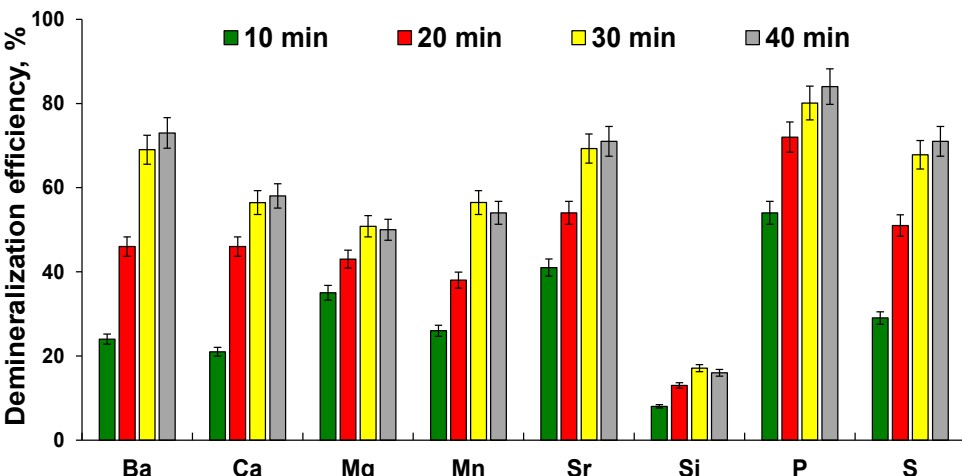

**Figure 5.** Demineralization efficiency with several extraction periods using an ultrasonic bath operating at 25 kHz. Experimental conditions: 250 mg of sugarcane straw and 20 mL of extraction solution 15% (v v$^{-1}$) at 70 °C and 70% of US amplitude (error bars are the standard deviation, n = 3).

Despite the parameters already evaluated in this section, which were not directly related to the US wave itself, the US amplitude brings information about its intensity. It is possible to find studies about solid-liquid extraction associating the US amplitude with high-efficiency metal and nonmetal extraction [24]. The amplitude is directly related to the ultrasound intensity, and the increase in the sonochemical effects with higher US intensities is expected. Consequently, the higher the US amplitude, the more effective the acoustic cavitation action, which intensifies the interfacial contact in solid-liquid extraction, increasing the action of the solvent and improving the elemental extraction [27–29].

In this sense, experiments were carried out to verify the most suitable US amplitude for the demineralization of sugarcane straw. The amplitudes ranged from 10 to 70%, as presented in Figure 6. As observed, 60 and 70% amplitudes were the most efficient for metal extraction (60 and 58%, respectively) when compared with lower US frequencies (from 24 to 45% of element extraction for US amplitudes ranging from 10 to 50%, respectively). Since no additional improvement in the elemental extraction was observed when 70% of US amplitude was used, 60% was selected as the most suitable condition for the extraction of metals and nonmetals.

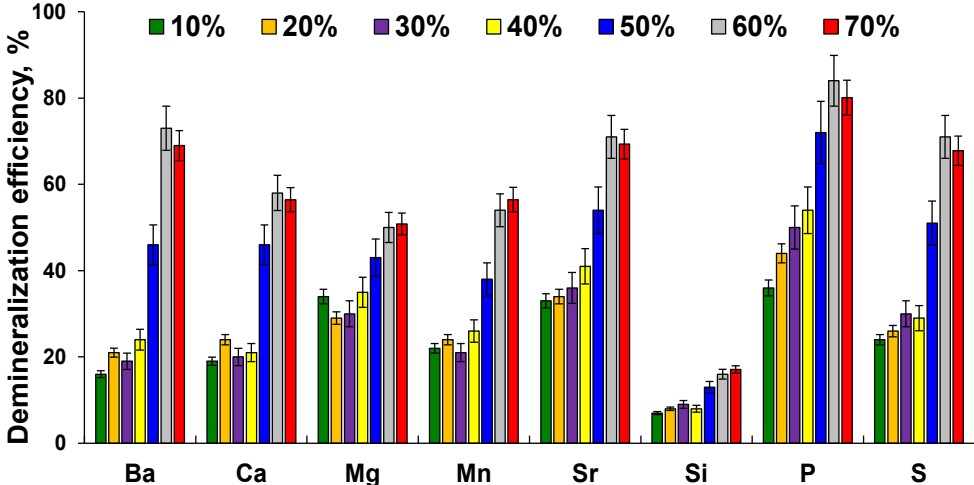

**Figure 6.** Demineralization efficiency with several US amplitudes using an ultrasonic bath operating at 25 kHz. Experimental conditions: 250 mg of sugarcane straw and 20 mL of extraction solution 15% (v v$^{-1}$) for 30 min at 70 °C (error bars are the standard deviation, n = 3).

### 3.4. The US Associated Effect: Comparison with Mechanical Stirring (MS)

To prove that US indeed contributed to extraction metals and nonmetals from sugarcane straw, the results obtained at optimized conditions were compared with those using MS (500 rpm) instead of US energy, at the same experimental conditions. As observed in Figure 7, for practically all the evaluated elements (except Mg), there was a significant improvement in the extraction efficiency (60 and 48% for UAD and process assisted by MS, respectively). One can assume that the better extraction efficiency when US is used, even compared with MS with relatively high rotation (500 rpm), is related to the acoustic cavitation. Acoustic cavitation can intensify the mass transfer in the solid-liquid extraction, which probably resulted in the intensification of the extraction process [29].

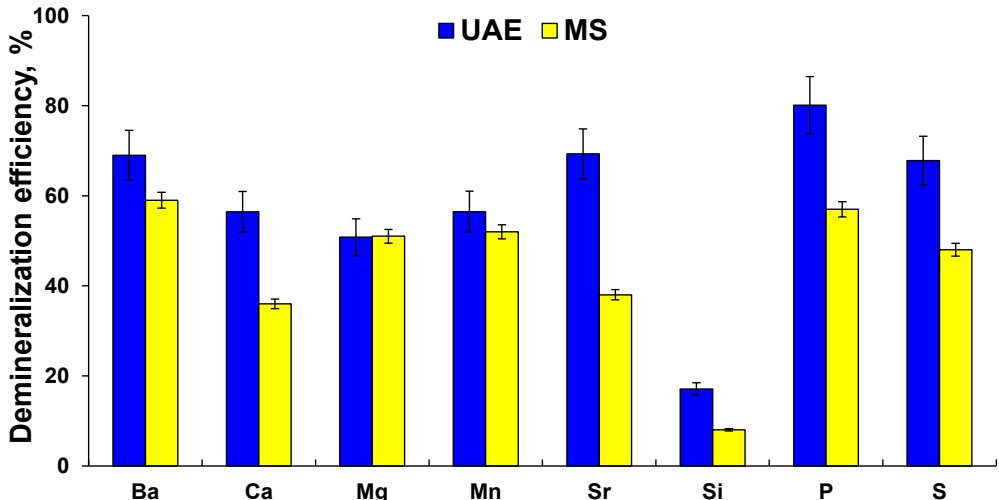

**Figure 7.** Demineralization efficiency using an US bath operating at 25 kHz and MS (500 rpm). Experimental conditions: 250 mg of sugarcane straw and 20 mL of extraction solution 15% (v v$^{-1}$) for 30 min at 70 °C and 70% of US amplitude (error bars are the standard deviation, n = 3).

### 3.5. Scale Up

To evaluate the possibility of applying the proposed processes in sugarcane straw demineralization, an attempt to scale the process up was performed. The conditions previously optimized for UAD were kept constant (15% $H_2O_2$ as the extraction solution, 30 min sonication time, extraction temperature of 70 °C, and 60% of US amplitude operating at 25 kHz). The solid–liquid relation was also kept constant, except the biomass amount, which was increased from 250 mg to 37 g, while the solvent increased form 20 mL to 3 L, as presented in Table 1.

As presented in Table 2, when the amount of biomass was increased up to 37 g (148-fold compared with the initial condition), while keeping constant the solid–liquid ratio, the extraction efficiency presented a slight improvement (from 60 to 66%, for the pool of the elements). It means that the proposed UAD process was possible to scale up, even considering a considerable reduction in the delivered power (from 184 to 57 W dm$^{-3}$ for 0.5 and 3 L loads, respectively). The US bath system used in this work had a maximum load of 3.5 L, which means that 3 L represent the working load for sugarcane straw demineralization. Even so, the extraction was not committed, showing the feasibility for scaling up the process.

**Table 2.** Experimental conditions and UAD efficiency for scaling up the process.

| Reactor Load (L) | Sugarcane Straw (g) | Power Density (W dm$^{-3}$) | Demineralization Efficiency (%) |
| --- | --- | --- | --- |
| 0.02 * | 0.250 | 115 | 60 |
| 0.5 | 6 | 184 | 61 |
| 1.0 | 12 | 172 | 64 |
| 3.0 | 37 | 57 | 66 |

* Experimental conditions used for the proposed UAD optimization.

### 3.6. SEM Images and the Morphological Evaluation of the Sugarcane Straw

Scanning electron microscopy (SEM) analysis provided information about the morphological structure of sugarcane straw. The images were collected before and after pretreatments with US energy, and MS.

As shown in Figure 8a, the untreated sugarcane straw had small orifices, typical of this kind of biomass. However, when the sample was subjected to US treatment, the holes slightly increased in diameter, and the fibers became more exposed (Figure 8b). The observed change in the biomass surface, although not so pronounced, was considered as evidence of the physical effects caused by acoustic cavitation, such as erosion, that results in deformations and disorganization of crystalline lignocellulosic structure, making the demineralization process easier [30].

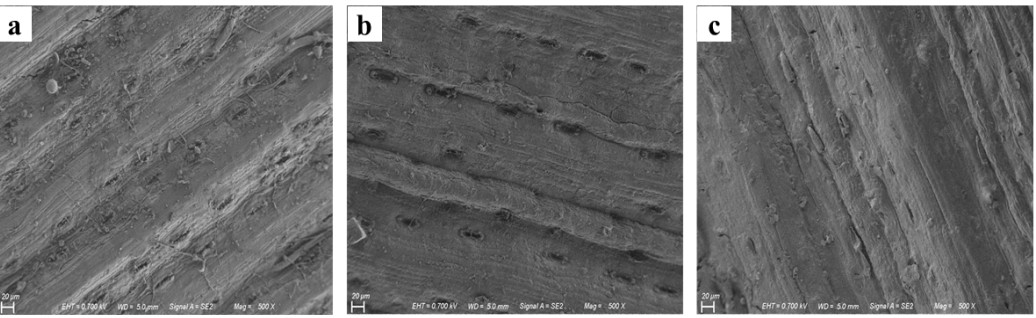

**Figure 8.** SEM images of: (**a**) untreated sugarcane straw; (**b**) sugarcane straw after treatment with US; and (**c**) sugarcane straw after treatment with mechanical stirring.

In Figure 8c, one can see that there were not so many changes in the morphological structure of sugarcane straw when compared to that of untreated biomass. This result corroborates the worst extraction efficiency when MS was used instead of US.

### 3.7. Influence of the Ultrasound-Assisted Demineralization on the Production of Carbohydrate Monomers by Acid Hydrolysis

Thermal conversion represents one of the main uses of sugarcane straw to generate energy, where the presence of metals can damage the pipelines/reactors due to in-crustation and/or corrosion, as well as reduce its calorific energy. The problems related to the presence of metals remain when considering the production of second-generation liquid biofuels (e.g., ethanol), which represents a growing trend related to the production of energy. In this way, to optimize an efficient pretreatment method for demineralization of sugarcane straw, it becomes important to evaluate how the UAD step would contribute to further biochemical conversion. Thus, the sugarcane straw (untreated, treated with US, and treated with MS) was subjected to acid hydrolysis.

The hydrolysis experiment was performed with 500 mg of sugarcane straw using dilute $H_2SO_4$ (8.3 mL of 1.5% solution). This step was carried out to reduce the biomass recalcitrance and to evaluate the influence of US energy and MS on the conversion of cellulose and hemicellulose from sugarcane straw into fermentable sugars, such as glucose, xylose, and arabinose. The results are presented in Table 3.

**Table 3.** Monomeric sugars' yield of sugarcane straw.

| Sugarcane Straw | Hydrolysate Concentration, mg L$^{-1}$ | | |
| --- | --- | --- | --- |
| | **Glucose** | **Xylose** | **Arabinose** |
| Untreated | 0.06 ± 0.01 | 2.65 ± 0.28 | 0.92 ± 0.15 |
| Ultrasound | 0.05 ± 0.01 | 4.10 ± 0.17 | 1.48 ± 0.12 |
| Mechanical stirring | 0.08 ± 0.01 | 4.11 ± 0.42 | 1.45 ± 0.13 |

The results observed from the hydrolysis of sugarcane straw clearly demonstrate that the production of fermentable sugars after the demineralization step with US or MS becomes easier. The total sugars' yield (glucose, xylose, and arabinose) increased up to 55% in both systems when compared with that from untreated biomass. Although there was no difference between the amount of carbohydrate monomers produced after pretreatment assisted by US or MS, the higher efficiency of UAD is desired for industrial purposes, since it reduces the change for corrosion and clogging of pipelines and reactors, as well as contributes to increasing the lifetime of the used catalysts.

## 4. Conclusions

Based on the results obtained in this study, it was observed that the US bath system achieved better extraction efficiency compared to that of mechanical stirring (60% and 42%, respectively), because of physical and chemical effects provided by ultrasound. It is worth mentioning that the use of a $H_2O_2$ 15% solution intensified metal removal, despite contributing to developing an environmentally friendly process, since water is the main residue. Additionally, the UAD as the pretreatment for the sugarcane straw enabled subsequent production of fermentable sugars, with improvement up to 55% in the final yield. Furthermore, scaling up was proposed, looking for a potential industrial application, which was well succeeded up to 3 L (up to 66% extraction efficiency). All these advantages demonstrate that UAD can be considered as an alternative process for metal removal from sugarcane straw.

**Author Contributions:** Conceptualization, C.A.B. and D.S.; methodology, D.S., C.M.S. and K.G.; software, D.S., C.M.S. and K.G.; validation, D.S., C.M.S., L.F.S. and K.G.; formal analysis, D.S., C.M.S., L.F.S. and K.G.; resources, C.A.B., E.M.M.F. and A.F.M.; data curation, C.A.B., D.S., C.M.S. and K.G.; writing—original draft preparation, C.A.B., D.S., C.M.S. and K.G.; writing—review and editing, C.A.B., C.M.S. and K.G.; visualization, C.A.B., D.S., C.M.S. and K.G.; supervision, C.A.B., E.M.M.F. and A.F.M.; project administration, C.A.B. and D.S.; funding acquisition, C.A.B., E.M.M.F. and A.F.M. All authors have read and agreed to the published version of the manuscript.

**Funding:** This research was funded by Coordenação de Aperfeiçoamento de Pessoal de Nível Superior–Brazil (CAPES)–Finance code 001.

**Acknowledgments:** The authors are grateful to Conselho Nacional de Desenvolvimento Científico e Tecnológico–Brazil (CNPq, Grant nr. 313786/2019-4; 309297/2016-8) and Fundação de Amparo à Pesquisa do Estado do Rio Grande do Sul–Brazil (FAPERGS, Grant nr. 17/2551-0000960-6; 17/2551-0000985-1) for supporting this study. This study was financed in part by the Coordenação de Aperfeiçoamento de Pessoal de Nível Superior–Brazil (CAPES)–Finance code 001.

**Conflicts of Interest:** The authors declare no conflict of interest.

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
