# Peer review of "Ultrasound-Assisted Demineralization Process of Sugarcane Straw and Its Influence on the Further Biomass Conversion"

_sustainability, doi:10.3390/su14010557_

Round 1

Reviewer 1 Report

The proposed study is an in-depth investigation on the use of ultrasound for biomass demineralization. There are some strong doubts that the authors need to answer.

The main question is: why  Ba, Ca, Mg, Mn, P, S, Si, and Sr were chosen?
The proposed technique would have been much more interesting if applied to elements of analytical interest (such as heavy metals). Authors should at least explain the scientific or environmental importance of extracting these specific elements.

Another unclear aspect is: why was hydrogen peroxide chosen? the authors declare to have tested different solvents in Fig.1 (HNO3, HCl, H2SO4, H2O2, and H2O), however it would have been scientifically more relevant to use mixtures of these as well. For example, it is known that in microwave mineralizers the best extraction performance for biomass is obtained with a solution of H2O2 + HNO3. Why haven't multiple solvent solutions been tested?

Other suggestions for the paper are:

In the abstract, instead of "liquid fuel" it is more appropriate "liquid Biofuel"

Line 59: "sugarcane" instead of "sugar cane"

In paragraphs 2.1 and 2.2 the authors say that "biomass was digested using a microwave-assisted sample preparation system" and list a number of solvents used for digestion. However, it would be advisable to specify the solvents, concentrations and methods used for the digestion of the samples

In 2.3 table 1 is a result obtained by the authors, it should be reported in the preliminary results paragraph

line 166: The authors speak of the analysis of sugars, but before now no mention has been made of why they are analyzed in this paper. Furthermore, the method used in HPLC should be described

3.1 Were the resulting solutions injected directly into the ICP or were the samples pretreated before? should be specified

Line 239 correct H2O2

3.6 SEM results are shown, but there is no hint of this analysis in the materials and methods

Line 384 Specify the dilution

The article is titled "Ultrasound-assisted demineralization process of sugarcane straw and its influence on the further biomass conversion" but it is never explained why the extraction of these elements should favor biomass conversion. Are we talking about energy conversion? and what kind? why subtracting these elements should improve the final product?

Author Response

The proposed study is an in-depth investigation on the use of ultrasound for biomass demineralization. There are some strong doubts that the authors need to answer.

Response: Authors thank you for the detailed revision of the manuscript, which was revised considering the Reviewer’s comments and suggestions.

1) The main question is: why Ba, Ca, Mg, Mn, P, S, Si, and Sr were chosen? The proposed technique would have been much more interesting if applied to elements of analytical interest (such as heavy metals). Authors should at least explain the scientific or environmental importance of extracting these specific elements.

R: This manuscript describes a prove of concept regarding demineralization process assisted by ultrasound. The elements were arbitrarily selected to prove that ultrasound energy can be used for improving the quality of biomass prior to its further conversion. Some elements, such as Ca, Mg, Mn, P, and S were selected because are present in large amounts in the sugarcane straw, while other (Ba, Si, and Sr) were present in relatively low concentration. All of them present distinct physical and chemical behavior, giving the possibility to estimate the behavior of other elements.

2) Another unclear aspect is: why was hydrogen peroxide chosen? the authors declare to have tested different solvents in Fig.1 (HNO3, HCl, H2SO4, H2O2, and H2O), however it would have been scientifically more relevant to use mixtures of these as well. For example, it is known that in microwave mineralizers the best extraction performance for biomass is obtained with a solution of H2O2 + HNO3. Why haven't multiple solvent solutions been tested?

R: Ideally, the demineralization process extract elements different from C and H. Some elements, such as N, S and Cl are well known to be catalyst poison. Thus, although the possibility of using mixtures of reagents would be interesting for increasing the demineralization process, they will certainly increase the extraction of C fractions, reducing the energetic power of biomass and the efficiency of sugar production.

The use of mixtures such as H2O2 + HNO3 increase the oxidant potential of HNO3, which are going to increase the fraction of C that becomes soluble, or even oxidized to CO2 (worst situation). The use of other reagents such as H2SO4, HNO3 or HCl might be considered source of catalyst poison; being better to be avoided if it is possible. Thus, the use of H2O2 was selected as the best condition, because it did not impair the demineralization efficiency, and resulted in environmentally friendly products (H2O and O2).

Other suggestions for the paper are:

3) In the abstract, instead of "liquid fuel" it is more appropriate "liquid Biofuel"

R: The term “liquid biofuel” was updated, as suggested.

4) Line 59: "sugarcane" instead of "sugar cane"

R: This term was checked along the text and corrected when necessary.

5) In paragraphs 2.1 and 2.2 the authors say that "biomass was digested using a microwave-assisted sample preparation system" and list a number of solvents used for digestion. However, it would be advisable to specify the solvents, concentrations and methods used for the digestion of the samples

R: The required information was added in item 2.4 Methods, since the authors believe in this section the required information becomes easier for searching and understanding.

“Microwave-assisted digestion of sugarcane straw was performed with a mixture of 5 mL of 14.4 mol L-1 HNO3 and 1 mL of 12 mol L-1 HCl in the following heating program: (1) 10 min ramp up to 200 ºC (2) hold for 40 min at 200 °C. The sample mass used in the elemental characterization was 300 mg. For the sugarcane straw digestion step after demineralization, the sample was separated from the extracting solution, dried and 200 mg of sample was digested. All the digests content in the vessels were diluted up to 20 mL with ultrapure water. For the ICP-OES analysis, the solutions resulting from the sample digestion were diluted with HNO3 5% (same condition as the calibration curves) and injected into the equipment.”

6) In 2.3 table 1 is a result obtained by the authors, it should be reported in the preliminary results paragraph

As suggested, the Table 1 is now reported at section 3.1 (Influence of extraction solution for demineralization process).

7) line 166: The authors speak of the analysis of sugars, but before now no mention has been made of why they are analyzed in this paper. Furthermore, the method used in HPLC should be described

R: Although the analysis of sugars was not the main target of the proposed study, it was used to evaluate how the pretreatment of sugarcane straw would contribute for further conversion of the biomass. To make it clearer, additional information was described in the introduction section (last paragraph).

The HPLC method used for sugar determination and quantification, was included in the section 2.4, as follows:

“The concentration of glucose, xylose, and arabinose after the hydrolysis process was determined by high performance liquid chromatography (HPLC) coupled to a refractive index (RID) detector. The chromatograph (Shimadzu, São Paulo, Brazil) was equipped with a LC-20AT pump, a communication module CBM- 20A, a DGU-20A5 degasser, and an autosampler SIL-20A that was used with the aid of LC Solution software. The separation was performed using an Aminex HPX-87H column (300mm×7.8 mm; 9.0 μm particles), at 50 °C, 0.5 mL min−1 flow (mobile phase 5 mmol L-1 H2SO4) in isocratic mode. Prior to chromatographic separation the samples were diluted in ultrapure water (1:10 v v-1) and, subsequently, passed through syringe filters (PTFE or MCE 13mm x 0.22 μm). The injection volume was 20 μL. The chromatographic run time corresponding to the analyte signal were 10.7 min to glucose, 11.5 min to xylose and 12.6 min arabinose.”

8) 3.1 Were the resulting solutions injected directly into the ICP or were the samples pretreated before? should be specified

R: The required information is present in section 2.4. After the extraction process with ultrasound, the solid residue from sugarcane straw was separated from the extraction solution, dried, and digested in a microwave oven (digestion method described in the section 2.4). The solution resulting from the digestion were diluted with 5% HNO3 and analyzed by ICP-OES.

9) Line 239 correct H2O2

R: It was corrected.

10) 3.6 SEM results are shown, but there is no hint of this analysis in the materials and methods

R: The method used for SEM analysis is described in the section 2.4.

11) Line 384 Specify the dilution

The used dilution (H2O2 15%) was specified, as suggested.

12) The article is titled "Ultrasound-assisted demineralization process of sugarcane straw and its influence on the further biomass conversion" but it is never explained why the extraction of these elements should favor biomass conversion. Are we talking about energy conversion? and what kind? why subtracting these elements should improve the final product?

R: The use of sugarcane straw to generate thermal energy represent the most common approach used for its biomass conversion. Considering this application, the presence of metals can damage the pipelines due to incrustation and corrosion and reduce calorific energy during biomass burning. However, considering that we are facing a growing trend related to the use of liquid biofuels, mainly those from second generation (e.g. ethanol from fermentation of lignocellulosic biomass), we consider appropriate to develop a pretreatment method able to remove metals, making easy further conversion process. In addition to that, the results demonstrate that after the treatment of sugarcane straw with ultrasound there was an increase in sugar yield, which is interesting for bioconversion processes for the generation of biofuels.

Reviewer 2 Report

This paper has attempted to develop the demineralization process for  lignocellulosic biomass wastes. The idea is good and the data were collected huge enough. Only one item shall be defined in more clear, the "demineralization process", and the related conceptual literature shall be cited. 

Author Response

This paper has attempted to develop the demineralization process for lignocellulosic biomass wastes. The idea is good and the data were collected huge enough. Only one item shall be defined in more clear, the "demineralization process", and the related conceptual literature shall be cited. 

Response: The authors appreciate your suggestion and time spent for revising this manuscript. The information related “demineralization process” was improved in the introduction section.

Reviewer 3 Report

This manuscript describes a method for removal of metals and non-metals from sugarcane straw using ultrasound energy. The authors examined the influence of several parameters including: solution type, H2O2 concentration, extraction temperature, sonication time and ultrasound amplitude on the demineralization performances. The introduction of ultrasound energy increase the demineralization efficiency up to 16%. The experimental results and interpretations are vague. I would like to recommend major revisions. Comments of this paper are shown below:

1) Page 6, Line 223. "When high US frequency from 45 to 130 kHz was used, extraction efficiencies from 37 to 50% were observed (130 and 45 kHz, respectively)." This statement does not agree with the data shown in Figure 2. Please clarify on this one.

2) Page 6, Line 232 to 233. The statement here is very vague. Please specify which element (or all element) you are analyzing, and try to correlate the statement to Figure 2 so that reader can understand.

3) Line 239, Please correct the format for "H2O2".

4) Line 241, Please correct the format for "dm-3"

5) Page 7, Line 251, "As observed in Figure 3, the extraction efficiency tends to increase from 5 to 251
15% H2O2 (32 to 57% respectively, for the pool of the elements)" This statement does not agree with the data shown in Figure 3. Please clarify which element you are referring to.

Author Response

This manuscript describes a method for removal of metals and non-metals from sugarcane straw using ultrasound energy. The authors examined the influence of several parameters including: solution type, H2O2 concentration, extraction temperature, sonication time and ultrasound amplitude on the demineralization performances. The introduction of ultrasound energy increase the demineralization efficiency up to 16%. The experimental results and interpretations are vague. I would like to recommend major revisions. Comments of this paper are shown below:

Response: The authors appreciate your suggestions and the time spent for revising this manuscript.

1) Page 6, Line 223. "When high US frequency from 45 to 130 kHz was used, extraction efficiencies from 37 to 50% were observed (130 and 45 kHz, respectively)." This statement does not agree with the data shown in Figure 2. Please clarify on this one.

It was a mistake, and the correction has already been made in the referred section.

2) Page 6, Line 232 to 233. The statement here is very vague. Please specify which element (or all element) you are analyzing, and try to correlate the statement to Figure 2 so that reader can understand. 251

R: the extraction efficiency refers to the total of elements (sum of all elements). The text has been updated for better understanding.

3) Line 239, Please correct the format for "H2O2".

R: The text has been revised and some typo were corrected.

4) Line 241, Please correct the format for "dm-3"

R: This typo was corrected.

5) Page 7, Line 251, "As observed in Figure 3, the extraction efficiency tends to increase from 5 to 15% H2O2 (32 to 57% respectively, for the pool of the elements)" This statement does not agree with the data shown in Figure 3. Please clarify which element you are referring to.

R: this trend is observed for practically all the evaluated elements (Sr and S were the only exceptions, as it is possible to see in Fig. 3). In the text, when the extraction efficiency is discussed, it is considered all the elements for a single experimental condition, which is better explained in experimental section (item 2.4).

Round 2

Reviewer 1 Report

The authors answered the questions exhaustively and corrected the paper appropriately.

However, I suggest to report in the paper, also summarizing, this concept: "The use of sugarcane straw to generate thermal energy represent the most common approach used for its biomass conversion. Considering this application, the presence of metals can damage the pipelines due to incrustation and corrosion and reduce calorific energy during biomass burning. However, considering that we are facing a growing trend related to the use of liquid biofuels, mainly those from second generation (eg ethanol from fermentation of lignocellulosic biomass), we consider appropriate to develop a pretreatment method able to remove metals, making easy further conversion process. In addition to that, the results demonstrate that after the treatment of sugarcane straw with ultrasound there was an increase in sugar yield, which is interesting for bioconversion processes for the generation of biofuels "

Author Response

The authors answered the questions exhaustively and corrected the paper appropriately.

However, I suggest to report in the paper, also summarizing, this concept: "The use of sugarcane straw to generate thermal energy represent the most common approach used for its biomass conversion. Considering this application, the presence of metals can damage the pipelines due to incrustation and corrosion and reduce calorific energy during biomass burning. However, considering that we are facing a growing trend related to the use of liquid biofuels, mainly those from second generation (eg ethanol from fermentation of lignocellulosic biomass), we consider appropriate to develop a pretreatment method able to remove metals, making easy further conversion process. In addition to that, the results demonstrate that after the treatment of sugarcane straw with ultrasound there was an increase in sugar yield, which is interesting for bioconversion processes for the generation of biofuels "

R: The authors appreciate your time and effort in improving the quality of our manuscript. The following sentence was included in section 3.7 (Influence of the ultrasound-assisted demineralization on the production of carbohydrate monomers by acid hydrolysis), as suggested. Thank you again.

“Thermal conversion represents one of the main uses of sugarcane straw to generate energy, where the presence of metals can damage the pipelines/reactors due to in-crustation and/or corrosion, as well as reduce its calorific energy. The problems related to the presence of metals remain when considering the production of second-generation liquid biofuels (e.g., ethanol), which represents a growing trend related to the production of energy. In this way, although to optimize an efficient pretreatment method for demineralization of sugarcane straw, it becomes important to evaluate how the UAD step would contribute to further biochemical conversion. Thus, the sugarcane straw (untreated, treated with US, and treated with MS) was subjected to acid hydrolysis.”

Reviewer 3 Report

The authors addressed most of my concerns in the new version of the manuscript, and the paper is appropriate for publication in its present form.

Author Response

Thank you again for your time and expertise, which were important for improving the manuscript quality.